# Monitoring Environmental Parameters with Oil and Gas Developments in the Permian Basin, USA

**DOI:** 10.3390/ijerph17114026

**Published:** 2020-06-05

**Authors:** Robert Nelson, Joonghyeok Heo

**Affiliations:** 1Staff Geologist, Larson and Associates, Midland, TX 79701, USA; nelson_r8014@utpb.edu; 2Department of Geosciences, University of Texas—Permian Basin, Odessa, TX 79762, USA

**Keywords:** groundwater quality, environmental change, Permian Basin, USA

## Abstract

This study evaluates the groundwater qualities and environmental changes to obtain information on the groundwater contamination in the Permian Basin, Texas. Coupled with the U.S. government’s open data, these analyses can identify regions where environmental change could have affected groundwater quality. A total of thirty-six wells were selected within the six counties: Andrews, Martin, Ector, Midland, Crane, and Upton. Spatial distribution maps were created for six different parameters: pH, total dissolved solids (TDS), chloride, fluoride, nitrate, and arsenic. Total groundwater quality maps incorporate all the contaminants and denote regions of poor, medium, and optimum conditions. To identify spatial changes in groundwater quality, maps were separated into two different time intervals, 1992–2005 and 2006–2019. We found that groundwater contamination resulted primarily from the mobilization of the contaminant from anthropogenic activities such as chemical fertilizers, oil and gas developments. Overall, groundwater quality decreased during the study period from 1992 to 2019 as population and urban growth began to develop in the Permian Basin. This study contributes on understanding of the response of groundwater quality associated with environmental change in the Permian Basin. Therefore, this research provides important information for groundwater managements in developing plans for the use of water resource in the future.

## 1. Introduction

Groundwater accounts for approximately 32 percent of all water supplied by municipal water treatment facilities. As population continues to rise, there will be a heavier reliance on this diminishing natural resource. The Safe Water Drinking Act (SWDA) was originally established by Congress in 1974, and developed groundwater quality regulations to protect over 150,000 public water systems across the U.S. [1]. Citizens obtaining their water from privately owned water wells in rural areas not serviced by local water treatment facilities should take preventative steps such as testing and decontamination prior to consumption, as these wells are not regulated by the Environmental Protection Agency (EPA). 

Once the EPA has determined that a contaminant poses a risk to water quality, they will develop a maximum contaminant level goal (MCLG). The maximum contaminant level goal is achieved when the most vulnerable individuals, such as infants, children, and the elderly, would not experience adverse health effects from exposure of the contaminant [2,3]. After the EPA has developed a MCLG for a contaminant, then the agency sets a maximum contaminant level (MCL). The EPA enforces MCL’s within the municipal water treatment facilities to maintain water quality but in order to enforce the MCL, they must be both economically and technologically viable [1,4]. The three parameters that are not controlled by the EPA in this study are chloride, pH, and total dissolved solids (TDS). The remaining three contaminants, arsenic, fluoride, and nitrate have specific MCLG/MCL values determined by the EPA. 

Numerous studies have been conducted to identify the effect of oil and gas production on environmental changes. Unconventional oil and gas production provides a decreasing trend of groundwater quality (chloride, nitrate) over time in the Permian Basin, West Texas [5,6]. The land-use changes in West and Central Texas during the shale boom of 2008–2012 are a direct result from the utilization for unconventional reservoirs and the development of energy resources and other human contributing activities [7,8]. Previous studies [5,6,7,8] have suggested that oil and gas development affected various entities, including groundwater quality and land-use changes. However, few studies have combined both groundwater and land-use factors to identify how oil and gas development related to the environmental changes. 

Some studies have been undertaken to estimate the groundwater quality according to oil and gas development in the US [9,10,11,12]. Various States, like California, Wisconsin, Ohio, Minnesota, Pennsylvania, Arkansas, and Colorado have had studies discussing the changes to groundwater quality since hydraulic fracturing had been introduced to each area. Long [9] studied some challenges for groundwater quality from the oil and gas industries in the States of Wisconsin and Minnesota. Even though it is challenging to determine the impact of oil and gas activities on groundwater aquifers, there are evidences that important parameters like pH and salinity are affected [10,11]. Thus, major ions of chloride (Cl) should be monitored to ensure groundwater quality. EPA also mention that levels of total dissolved solids (TDS) can be affected by hydraulic fracturing practices [12].

Recent advances in oil recovery from unconventional reservoirs have drastically increased oil production operations in the Permian Basin [13,14,15]. The area in which this research is conducted is generally utilized for ranching, agriculture, and oil and gas production [14]. Due to the importance of groundwater in the production of these valuable commodities, maintaining stable groundwater is a necessity. As a result, a sharp increase in population and urban growth in west Texas has altered the landscape, potentially changing groundwater quality [15]. Therefore, further research on the Permian Basin, West Texas must be conducted in order to obtain a better understanding of the effects of groundwater quality over time.

The purposes of this research are to (1) describe an overview of current groundwater quality in the Permian Basin, (2) determine spatial distribution of groundwater quality parameters such as pH, TDS, chloride, fluoride, nitrate, and arsenic concentrations, and (3) provide total groundwater quality and environmental change maps from 1992 to 2019 in the study area. This research contributes to understanding of the responses of groundwater resources in the Permian Basin, Texas. Thus, this research can provide important information for groundwater resources manager in making decision and developing plans for use of the groundwater resources in the future.

## 2. Study Area

The study area is located in Western Texas which has a total area of 538.98 km^2^. It extends across six counties of Texas: Andrews, Martin, Ector, Midland, Crane, and Upton (Figure 1). The land cover of the study area consists mostly of developed, barren, bush, grass, and crop. The alluvial environment in which the sediments were deposited consisted of interbedded sand, silt, clay, and gravel filling prehistoric river valleys [16]. Deposition of this aquifer began during the late Miocene to the early Pliocene and formed from eastward flowing streams originating from the Rocky Mountains [13].

Groundwater originating from within the study area is captured from four aquifers: Ogallala (major), Pecos Valley (major), Edwards Trinity Plateau (major), and Dockum (minor). The Ogallala aquifer is the largest aquifer in the United States and is a major aquifer of Texas, underlying much of the High Plains region. It consists of sand, gravel, clay, and silt and has a maximum thickness of 800 feet. The Pecos Valley aquifer is among the major aquifers in West Texas. Water-bearing sediments include alluvial and windblown deposits in the Pecos River Valley. The Edwards-Trinity Plateau aquifer is a major aquifer extending across much of the southwestern part of the state. Water quality ranges from fresh to slightly saline, and most of the groundwater is used for irrigation, municipal supplies, industrial use, and power generation. The Dockum aquifer is a minor aquifer found in the northwest part of the state. It is a sandstone aquifer and the basal member of the Dockum formation with the upper layers being predominantly siltstone and claystone.

These aquifers are a valuable source of water for ranchers, farmers, and the recovery of oil and gas in the region. The deepest groundwater well is within the Dockum at 1600 feet and the Ogallala contains the shallowest at 70 feet (Figure 2). Recharge of the aquifer occurs primarily through infiltration of precipitation. Due to the high rate of evaporation in this arid region, very little reaches the water table. The recharge rate of this aquifer is lower than the depletion rate with variations from state to state. The study area in the Permian Basin, West Texas is currently experiencing the highest depletion rate, whereas certain areas have seen a drawdown of as much as 100 feet [17]. As for the hydraulic characteristics, average transmissibility are 365 feet^2^/day, and the storage coefficient was 0.074. Lateral movement of ground water from the Ogallala likely occurs in the northern edge of the region. Hydraulic fracturing is also known to affect groundwater reservoirs. During hydraulic fracturing, various chemicals are injected underground site to generate fractures and increase the production of hydrocarbons in reservoirs that have low-porosity and low-permeability [6].

The study area is categorized as a semi-arid climate, where temperatures can drastically fluctuate throughout the day. The Permian Basin average low temperatures for January are 28°F and July high temperatures are 95°F [17]. The region receives on average 13–18 inches of rain annually, mostly during the spring (March–May) and early fall months (September–October). During the late summer and early fall months, moist air originating from the tropics begins to rise due to the southwestern monsoon which is the primary producer for rainfall events in west Texas [18]. With this low average rainfall, the evaporation rate is greater than the precipitation rate, resulting in a dry climate with relatively low humidity. 

## 3. Data and Methods

### 3.1. Groundwater Quality Parameters 

All groundwater data were obtained from the Texas Water Development Board (TWDB)—Groundwater Database (GWDB), which is an open database provided by the US government. We collected all historically available data in our study area from TWDB-GWDB and checked the location of their wells. To apply the interpolation analysis with Geographic Information System (GIS, ESRI® Developer Network, CA, USA) and evaluate the spatial distribution of groundwater qualities, the observed wells must be uniformly distributed throughout the study area [7,19,20,21]. For this reason, we finalized the total 36 wells to create an even distribution throughout the study area (Table 1). 

For comparing historical changes in groundwater quality, data from 1992–2005 were correlated with data from 2006–2019. The groundwater quality parameters used for this study include pH, and total dissolved solid (TDS), chloride (Cl), fluoride (F), nitrate (NO_3_^−^), and arsenic (As). The standard quality for water has been specified by EPA and World Health Organization (WHO) (Table 2). 

The selected groundwater parameters data in the study area were required to provide long-term data with a relatively dense hydrological observation network. Each well was mapped within the study area using its provided latitude/longitude coordinate and contaminant concentration level. Nas and Berktay [7] analyzed six groundwater parameters (pH, conductivity, chloride, sulfate, hardness, and nitrate) in an urban area of Turkey and applied spatial analysis to evaluate the spatial distributions of the groundwater parameters using ArcGIS software from Environmental Systems Research Institute (ESRI). They finally generated the map of the total groundwater quality that was produced by overlapping of the six groundwater parameters. We followed and developed the methodology from Nas and Berktay [7].

Spline interpolation was implemented to evaluate the spatial distribution of groundwater concentrations in the map. The spline interpolation estimates values using a mathematical function that minimizes overall surface curvature, resulting in a smooth surface for the study area [21]. Once the groundwater maps were created for each different parameter, a map indicating total groundwater quality was produced. According to Nas and Berktay [7], the mosaic to new raster tool was performed by merging the six groundwater maps together while also adding all the variables. The total groundwater map from poor to optimum conditions were classified to denote regions with increasing and decreasing water quality [7,19]. The map represents a combination of the six groundwater parameters.

### 3.2. Examining the Environmental Change

The land-cover data were obtained from Texas Natural Resources Information System (TNRIS). TNRIS provided the National Land Cover Data (NLCD) for 1992 and 2011 in a raster format. The data have spatial resolution at 30 m for a single pixel measures 30 m in width and length, which is 900 m^2^. Raster data management was preformed to merge these raster data sets together.

For both images to fit inside the study area, an extract by mask process was performed. This process extracts the raster cells of the imagery data and places them within the county boundaries of the study area. Additionally, we applied spatial analyses to refine the data and identify issues including edge effect, resolution change artifacts, and misclassification (Figure 3). A majority filter was selected to smooth spatial anomalies and to provide a smoother and clearer image [21].

This research utilized seven different geographic features to describe variations from 1992 to 2011, which include water, developed, barren, forest, bush, grass, and crop (Table 3). In order to correlate between the NLCD 1992 and 2011, a reclassification and grouping of the land-cover was required [22,23]. Developed classification grouped the open space, low, medium, and high intensities, such as urban settlement, transportation and industrial land. Barren classification included the bare land, rock, sand, and clay. The forest category grouped the deciduous, evergreen, and mixed forests. The bush category grouped the shrub and scrub, which are less than six meters high. The crop category included the pasture, hay, cultivated crop. The wetland classification was removed from the 1992 and 2011 maps as their relevancy within this arid region is negligible. 

## 4. Results and Discussion

### 4.1. pH

Municipal water treatment facilities must regulate and balance pH prior to its distribution in order to provide optimal water. The concentration of hydrogen (H^+^) and hydroxyl (OH^−^) ions in a liquid determine the measurement of pH. These measurements decide whether the liquid is an acid, neutral, or alkaline. For this study, it is surmised that soil pH and groundwater pH have comparable concentration levels. Within soils, pH controls the mobility of contaminants such as arsenic and fluoride. As soils become acidic, fluoride mobility increases allowing concentrations to increase [24]. Alternatively, as pH shifts becoming alkaline, arsenic mobility increases, resulting in elevated concentrations [25]. These fluctuations in pH control the concentration levels of arsenic and fluoride in groundwater and soils. 

The pH in the study area ranges from 6.7–8.1 SU (Figure 4), where the deeper wells are more alkaline than the shallower wells. This increase is a result of the aquifers composition, where dissolving limestones and dolomite minerals contribute to the aquifers alkalinity. Corrosive water (pH lower than 7) has potentially damaging effects on the municipal water treatment facility and the local homeowner. These waters have the potential to corrode pipes and destroy household appliances. The EPA recommends well waters be between 6.5–8.5 SU to prevent these damaging effects on infrastructure [26]. Soil pH is affected by climate, temperature, and parent material [27]. In arid climates, the low precipitation results in soils that are closer to neutral or slightly alkaline due to the weathering and leaching effects of rainfall. The weathering of parent material results in the formation of soil horizons and is a contributing factor to the soils pH.

### 4.2. TDS

Total dissolved solids (TDS) is a measurement of the dissolved combined content of inorganic salts and small amounts of organic matter that are dissolved in water [3]. These inorganic salts can contaminate the groundwater through anthropogenic or natural activities. Increased agricultural activity can result in higher concentrations of total dissolved solids. The adverse effects of TDS on infrastructure and taste can occur due to increased concentrations (1000 mg/L or greater) in groundwater [26]. To prevent these effects from occurring, while also providing an acceptable taste to the user, it is recommended that the concentration of TDS does not exceed 600 mg/L [28].

As crop land began to develop within Martin County from 1992 to 2011, the concentration of TDS continued to rise. Between 1992–2005 and 2006–2019, concentration levels rose from 1140 mg/L to 5000 mg/L (Figure 5). Elevated TDS levels can result from plants’ uptake of water, allowing soil to retain chlorides [28]. When additional precipitation or irrigation is contributed to the soil, these chlorides can leach through the subsurface and reach the underlying groundwater. Furthermore, agricultural activities as discussed in τ 4.8. (Environmental Change), environmental changes, chemical fertilizers and concentrated animal feeding operations, contribute to the increase TDS levels observed. 

The disposal of oilfield brines resulted in the higher concentrations of TDS levels in northwest Ector County. Concentrations rose from 27.3 mg/L in 1992–2005 to 3322.5 mg/L in 2006–2019 (Figure 5). The developed land-cover change from 1992 to 2011 is noticeable in this area from the development of production pads and saltwater disposal sites (Figure 12). TDS levels may also rise from natural weathering of rocks and soils in the subsurface. Regions with limestone and dolomite present can have naturally high levels of TDS due to the presence of calcium.

### 4.3. Chloride

The chloride contamination of groundwater and the water supply have the potential to threaten the environment. The consumer may notice the prevalence of the chloride anion within drinking water at higher concentrations, producing a salty taste. The taste thresholds can range from 200 to 1000 mg/L and are determined by the associated cation of either sodium, calcium, or magnesium [3]. Municipal water treatment facilities require sodium chloride concentrations to remain below 250 mg/L to provide optimal water quality and prevent the bitter taste that the consumer may detect [29]. Additionally, calcium or magnesium chloride concentrations may not be detected by the consumer until they reach levels of 1000 mg/L or greater. While chlorides in and of themselves pose little threat to human health, when paired with the cation sodium, both heart and kidney diseases may arise.

Between the two separate time intervals, chloride concentrations have remained relatively low except for in three locations: Martin County, Southern Crane County and Northwestern Ector County. In Martin County, chloride levels have risen from 406 mg/L in 1992–2005 to 2400 mg/L in 2006–2019 (Figure 6). This dramatic increase in concentrations is attributed to the development of additional farmland and the pollution factors associated with the production of cultivated crops. Furthermore, the depth of the underlying Ogallala aquifer in this region is shallow, allowing for chlorides to penetrate through the subsurface easily. In both Southern Crane County and Northwestern Ector County, chloride concentrations have remained high between the two separate time intervals. Both locations have a shallow depth of their underlying aquifer, making them susceptible to contamination.

Fluids associated with the production of oil and gas commonly consist of sodium chloride (saline/brines) and contain increased concentrations of total dissolved solids [30]. These fluids also consist of heavy metals and Naturally Occurring Radioactive Material (NORM), specifically Ra^226^ from the natural decay of U^238^ in shales and must be properly disposed to prevent the degradation of the municipal water treatment facility. The contamination of groundwater with oilfield brines is predominantly associated with surface spills, where fluids can reach shallow groundwaters due to leaking tanks, flowline ruptures, or other oilfield mechanical failures [6,30]. To determine whether the groundwater has been contaminated from a spill, chloride delineations must be obtained to identify the depth that the brines have penetrated through the subsurface.

### 4.4. Fluoride

Fluoride (F) is essential for the maintenance and solidification of our bones and to prevent dental decay. It has beneficial effects on teeth and bones when it is present at low concentration in drinking water. Fluoride in water keeps teeth strong and reduces cavities by about 25% in children and adults. However, it may cause mottling of the teeth depending on the concentration, the age of the child, the amount of drinking water consumed, and the susceptibility of the individual [31]. The presence of fluorides in groundwater is most frequently associated with weathering, where water passing through the subsurface encounters fluorine bearing minerals [32]. Through environmental studies of fluoride in the subsurface, there is a direct correlation between mobility of pH and fluoride [24]. Soils that are more acidic allow for an increase in fluoride mobility and leaching, where plant roots may subsequently accumulate the additional fluoride. This characteristic is demonstrated within the agricultural fields of southeast Martin County (Figure 7). In this region, there is a noticeable decrease in the concentration of fluoride between the two different time intervals.

Land-cover change from 1992 to 2011 indicates a substantial transformation of grass and bush to an area dominated by cultivated crops (Figure 12). As land turned to crops and the pH became acidic, plants began to uptake additional fluoride decreasing the concentration of the contaminant within groundwater. Fluoride concentration levels decreased from 7.8 mg/L in 1992–2005 to 3.35 mg/L in 2006–2019. Regions consisting of a lower annual rate of precipitation generally have a higher concentration of fluoride than regions with a higher annual rate of precipitation [31]. The average annual precipitation within the study area ranges from 13–18 inches, potentially increasing the concentration of fluorides in groundwater. Additionally, “residence time”, or the time it takes for groundwater to reach a well or stream, determines contaminant concentrations. In unconfined aquifers such as those of the Ogallala, Edwards-Trinity Plateau, and Pecos Valley, residence time can range from days to years. This swift migration of groundwater contributes to the overall decline in fluoride concentrations. Whereas in confined aquifers such as the Dockum, groundwater residence time can range from centuries to millennia, preventing rapid removal of contaminants.

While fluoride concentrations have remained relatively consistent throughout the study area, one well (ID: 4520202) within southwest Ector County accounts for the highest accumulation of fluorides. Concentration limits have remained relatively consistent from 8.27 mg/L in 1992–2005 and 8.55 mg/L in 2006–2019, well above the MCL of 4.0 MG/L. Located at the site of this well is a cement production facility (Figure 8). Fluorspar (calcium fluoride) is a common mineralizing agent in the manufacturing of cement [33]. In the oilfield, casing cement is used to isolate oil, gas, and water zones from the wellbore while also bonding the casing to the wall. Runoff and infiltration from the production of these materials has resulted in the elevated readings observed. Furthermore, this well passes through the confined Dockum aquifer, where groundwater has an increased residence time and natural attenuation of the contaminant is lower. 

### 4.5. Nitrate

Nitrate (NO_3_^−^) is a naturally occurring compound which sustains healthy plants in the ecosystem. It is in a more stable oxidation state than nitrite (NO_2_^−^) due to its extra oxygen, resulting in nitrite being detected with increased concentrations in a reducing environment [34]. While nitrates are found naturally occurring in groundwater, elevated concentrations are typically the result of anthropogenic activities [35,36]. Contamination of the groundwater from nitrates is frequently associated with the infiltration of inorganic nitrogen fertilizers and livestock waste from agricultural procedures [36]. Nitrogen fertilizers are commonly applied annually to increase the overall quality of cultivated crops and increase output. These fertilizers can reach the groundwater through nitrate leaching in the subsurface. 

The Environmental Protection Agency (EPA) has established a specific MCL for nitrate at 10 mg/L and elevated concentrations pose a small threat to human health, particularly in infants [35]. Groundwater wells rarely exceed this limit unless located in regions with increased agricultural activity. Well depth is an underlying factor affecting the concentration levels of nitrate in groundwater. As well depth increases, nitrate levels decrease, whereas shallower wells are observed to have the highest concentrations. This results from the leachate’s ability to penetrate through the subsurface easier and reach the underlying aquifer. Groundwater wells drawing from the deepest Dockum aquifer had the lowest levels, whereas the shallower Ogallala contained elevated nitrate concentrations.

From 1992–2005 the highest nitrate levels were observed in Northwest Andrews County, east Martin County, and south of the city of Midland. Between the two separate time intervals, there was a noticeable shift in concentrations (Figure 9). The concentrated animal feeding operation in northwest Andrews County experienced a reduction in nitrate levels, potentially from a decrease in operations or through better management processes. The percentage of crop increased within the study area, most notably in Eastern Martin County (Figure 12). Coupled with a shallow groundwater well depth, this resulted in nitrate levels continually increasing over time in this region. Furthermore, as the city of Midland continuously experienced urban growth, the application of inorganic nitrate fertilizers on household lawns and public parks elevated nitrate concentrations [37].

### 4.6. Arsenic

Arsenic is naturally found within the subsurface as a trace element on rocks or soils and is commonly used in agriculture activities. The two valence states of arsenic often found in groundwater are arsenite (As^+3^) and arsenate (As^5+^), but concentrations rarely exceed the recommended EPA MCL of 10 µg/L [2,38]. Arsenic poisoning (arsenicosis) can occur from exposure of 50 µg/L or greater contaminated groundwater and can lead to harmful effects on the human body such as an increased risk of cancer, diabetes, and damage to internal organs [6,25]. Within alkaline environments (pH greater than 7), arsenic becomes mobilized allowing for increased concentrations to be observed. These environments promote the release of arsenic through the electrostatic repulsion of the negatively charged Fe oxides/hydroxides and arsenic compounds [11,25]. 

The unconfined Ogallala aquifer contains the highest concentrations of arsenic where levels reach 45.5 µg/L (2006–2019) in northeast Martin County (Figure 10). Between the two separate time intervals, pH begins to shift to becoming more acidic within Northeast Midland County and Southeast Martin County. This shift resulted in a decrease in the mobility of pH and disrupted further release of arsenic in groundwater. Water quality improved between the two different time intervals within the city of Midland from 23.87 µg/L in 1992–2005 to 17.02 µg/L in 2006-2019. Furthermore, as pH or alkalinity increased in Northeast Martin County, the concentration levels remained consistent, resulting from mobility of arsenic in the subsurface. 

Other potential inputs of arsenic in groundwater occur when petroleum hydrocarbon releases create reducing environments allowing for its mobilization [39]. Microbial activity increases the degradation of the hydrocarbons and consumes terminal electron acceptors producing these environments. Once the microbial activity has progressed and there is a reduction in the redox conditions, the concentration of the arsenic in groundwater will decrease and return to its ambient levels [39,40]. While exposure to high concentrations can be fatal, smaller concentrations (8–14 µg/L) can also lead to damaging effects such as skin lesions [38]. It is important for individuals obtaining drinking water from wells to regularly test and purify for arsenic to prevent these damaging effects. 

### 4.7. Total Groundwater Quality

Water table can change over time due to changes in precipitation patterns, streamflow amount, and human-induced changes such as groundwater pumping and land development [11,41]. Changes in water table in wells are driven by the interplay between groundwater recharge and discharge to and from aquifers. In general, water tables in wells decline due to increased groundwater withdrawal and/or reduced aquifer recharge. The risk of contamination is greater for unconfined (water-table) aquifers than for confined aquifers because they usually are nearer to the land surface, and they lack an overlying confining layer to impeding the movement of contaminants. Because groundwater moves slowly in the subsurface and many contaminants sorb to the sediments, the restoration of a contaminated aquifer is difficult. In unconfined aquifers, contaminants from the soil or subsurface will directly affect the groundwater quality.

A wide range of different chemicals can be dissolved in groundwater as a result of interactions with the atmosphere, the surficial environment, soil and bedrock. The mutual influence of various chemical factors helps to evaluate hydrological processes responsible for changes in the groundwater quality. Groundwater tends to have much higher concentrations of most constituents than the surface waters do, and deep groundwater that has been in contact with the rock for a long time tends to have higher concentrations of the constituents than the shallow water. Shallow groundwater consists of Ca (calcium)–Na (sodium)–HCO_3_ (bicarbonate) dominantly formed by the interaction between atmospherically recharged meteoric water with the soil and shallow bedrock. These waters are usually fresh but upwelling of deeper saline fluids or saline intrusions from adjacent seawater bodies can influence their chemical composition [41]. Intermediate or deep depths groundwater rapidly increase in concentration of constituents primarily by the addition of SO_4_ (sulfate) and Cl (chloride). The concentration of bicarbonate ions decreases because of the precipitation of mineral phases such as calcite. Local variations in chemistry and anions may be due to a variety of rock–water interactions or local processes that result in Na–SO_4_, Na–HCO_3_, and Mg–SO_4_ type waters. The pH begins to rise in this zone and oxygen-consuming reactions and redox mineral controls tend to lower the Eh [42].

Groundwater quality maps were created using a modified version first discussed by Ducci [19], where thematic maps were initially produced using the interpolation method and subsequently stacked developing the total groundwater quality map [7,19,24]. The modified version utilizes a summation process, combining and calculating the six contaminants together, and defining regions of poor, medium, and optimum groundwater conditions (Table 4).

While not all contaminants have the same unit of measurement, the conversion of micrograms in arsenic to milligrams would result in a variable that is insignificant to the overall groundwater quality. Furthermore, the effects of pH are minimal on the total maps, as this variable has small fluctuations from 6.7–8.1.

The maps of total groundwater qualities demonstrate the changes that have occurred in 1992–2005 and 2006–2019 (Figure 11), identifying areas where additional environmental studies or regulatory practices could be implemented to improve overall water quality. Between the two separate time intervals, there is a moderate decrease in the amount of optimum groundwater quality within the study area. The two regions of notably poor quality are in Eastern Martin County and Northwest Ector County. This decrease in overall groundwater quality is attributed to the elevated levels of TDS and chlorides associated through anthropogenic activities. Within the city of Midland, groundwater quality has remained consistent throughout the time intervals with medium contaminant concentrations being observed. Odessa has experienced a slight decrease in groundwater quality due to the result of population growth and urban development. Table 5 showed the average of all different parameters in the study area for (a) 1992–2005 and (b) 2006–2019.

### 4.8. Environmental Change

With the advancement of horizontal drilling and hydraulic fracturing, previously untouched shale strata were now viable, greatly increasing the amount of recoverable reserves. As a result, rig count dramatically rose to nearly 500 at the end of 2011 and oil prices peaked at over USD 100 per barrel, ultimately allowing production to reach 1 million barrels per day [43]. Additionally, associated oilfield facilities grew to account for the increase in production. These factors resulted in a considerable change in both economic growth and land cover from 1992 to 2011 where the amount of developed land increased by 11.78 km^2^ or 176%, while also decreasing the percentage of barren and grass throughout the study area (Table 6).

Furthermore, the shale boom resulted in a noticeable increase in well sites and production pads scattered throughout West Texas, contributing to the growth of Developed land (Figure 12). The population within Midland rose from 89,443 in 1990 to 111,147 in 2010, while Odessa expanded from 89,699 in 1990 to 99,940 in 2010 [44]. Economic benefits associated with horizontal drilling and hydraulic fracturing such as higher incomes and extra jobs resulted in this flux of population growth. The increase within these two cities from 1992 to 2011 is depicted with a denser packing of the developed land.

Technological advances throughout the 1990s and 2000s allowed for an increase in agricultural output and quality. Global Positioning Systems (GPS) continually improved throughout the 1990s and into the 2000s, resulting in automated farm equipment that efficiently map and plan fields, increase production, and reduce the overall price from seed to harvest. Within this arid region, the primary water source is the Ogallala aquifer, which provides the necessary groundwater vital to produce various crops. The development of center pivot irrigation in the 1950s supported farmers to establish crops in these arid regions. Furthermore, genetic engineering of crops in the mid to late 1990s, lead to plants that are more resistant to insects, weeds, and viruses [45]. These factors, coupled with a growing demand from a rising population, allowed the percentage of Crop from 1992 to 2011 to increase by 14%, primarily within Martin County. To protect groundwater quality from further human activities, government agencies and local communities should adopt the following strategies: regulating the disposal of produced fluids from oil and gas activities, controlling the amount and type of agricultural chemicals, promoting the responsible use of waste drainages in the vicinity of mining areas, and boosting the development of rural and industrial infrastructures.

## 5. Conclusions

We evaluated groundwater quality parameters such as pH, TDS, chloride, fluoride, nitrate, and arsenic from 1992–2005 and 2006–2019 and identified land cover maps where specific changes in the environment effected groundwater quality in the Permian Basin, Texas. Utilizing advanced geospatial techniques, these parameters described areas from optimum and poor groundwater conditions. Factors that contribute to the level of contaminants in groundwater are natural sources, anthropogenic activities, and aquifer depth. The mobilization of arsenic and fluoride from natural sources resulted primarily in the fluctuations of the subsurface pH. These alterations in pH directly resulted in varying concentrations of contaminates within the groundwater. Anthropogenic activities such as petroleum spills and inorganic chemical fertilizers contributed to the contaminant load in groundwater. An underlying factor for the contamination is aquifer depth, where contaminants may reach shallower unconfined aquifers quickly, yet have a decrease in residence time. Developed and crop land cover significantly rose from NLCD 1992 to NLCD 2011 due to the increase in production from unconventional natural resources and advances in crop management. These increases consequently resulted in a decrease in overall groundwater quality in the Permian Basin, Texas. Total groundwater quality maps demonstrate a transition of water quality related to the advancement of urban development and population. This research provides significant information for the management of groundwater resources and the response to these potential changes in the Permian Basin, Texas.

## Figures and Tables

**Figure 1 ijerph-17-04026-f001:**
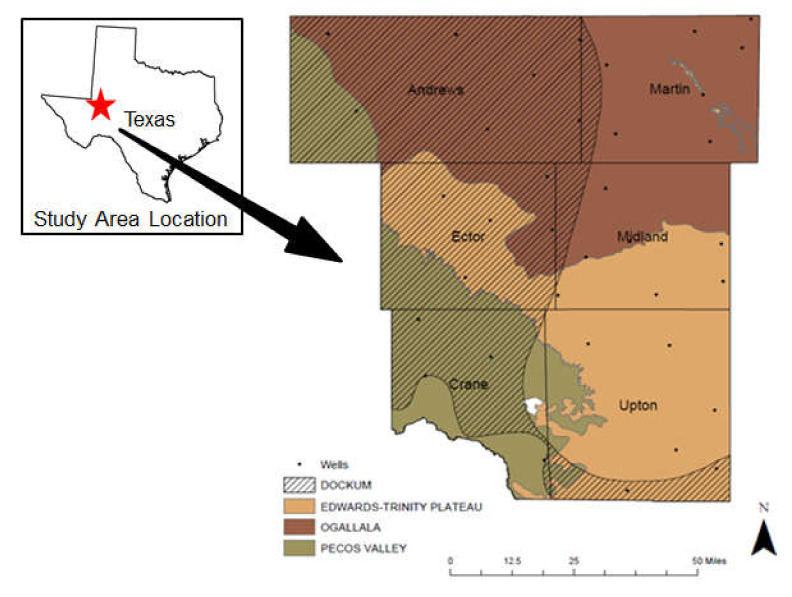
Study area map showing well locations, county and aquifer names.

**Figure 2 ijerph-17-04026-f002:**
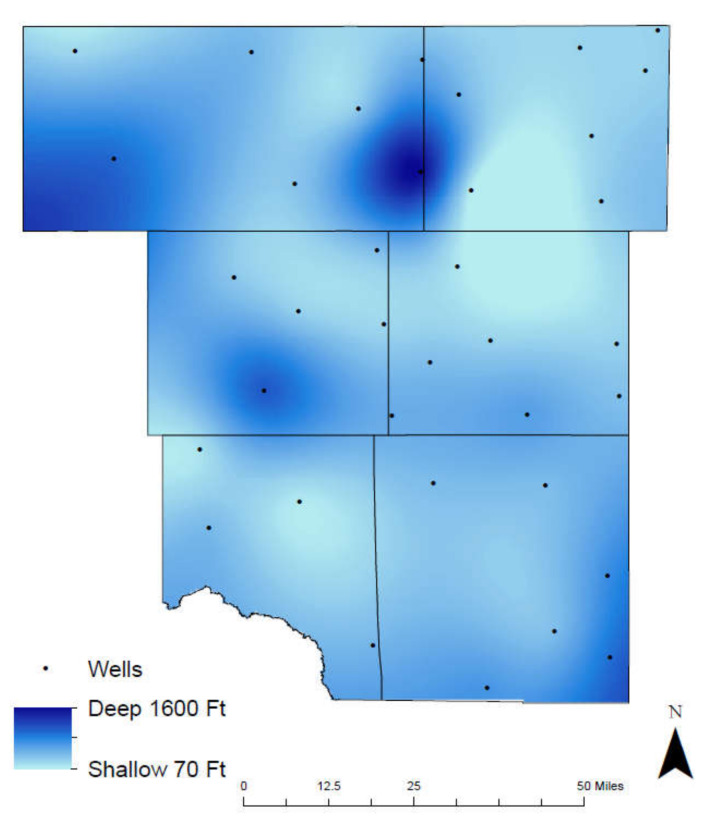
Groundwater well depth in the study area.

**Figure 3 ijerph-17-04026-f003:**
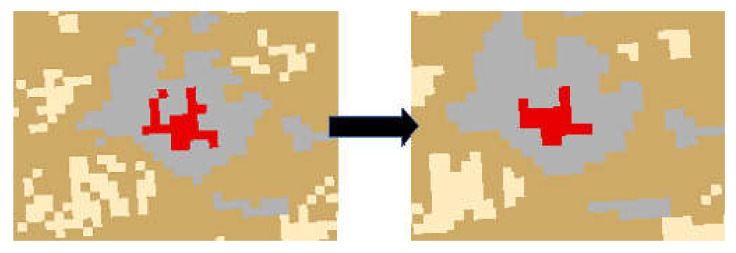
Example of majority filter in the study area.

**Figure 4 ijerph-17-04026-f004:**
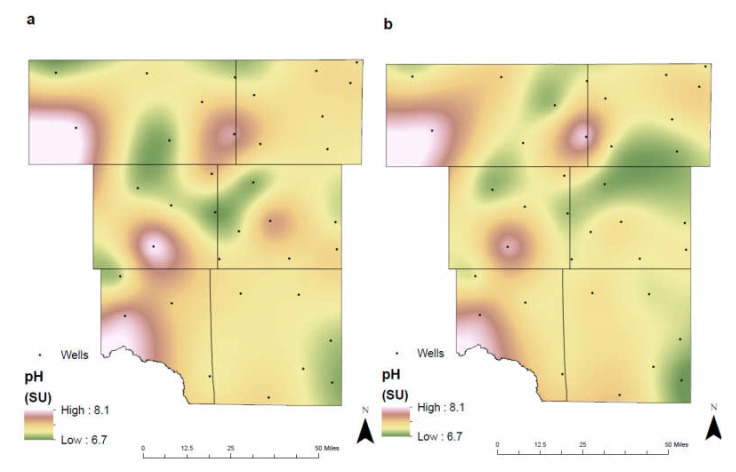
Spatial distribution of pH in the study area for (**a**) 1992–2005 and (**b**) 2006–2019 (SU: standard units).

**Figure 5 ijerph-17-04026-f005:**
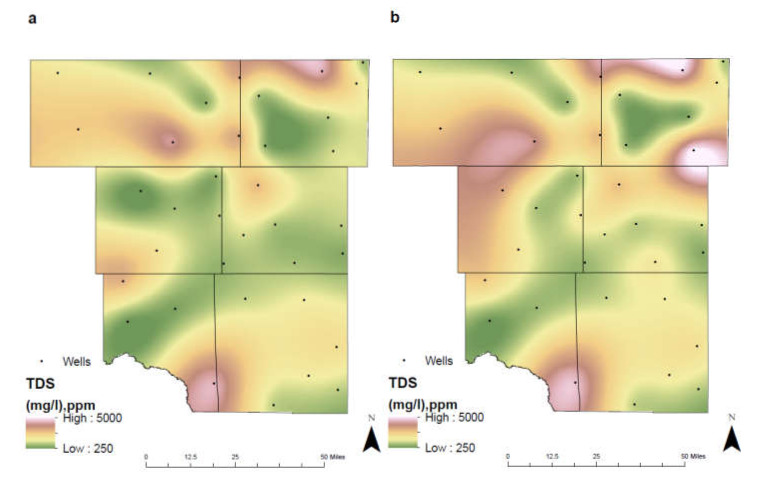
Spatial distribution of TDS in the study area for (**a**) 1992–2005 and (**b**) 2006–2019.

**Figure 6 ijerph-17-04026-f006:**
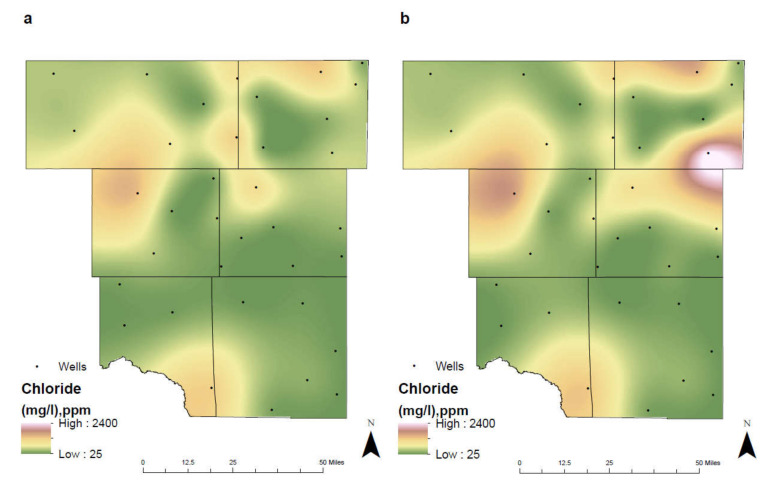
Spatial distribution of chloride in the study area for (**a**) 1992–2005 and (**b**) 2006–2019.

**Figure 7 ijerph-17-04026-f007:**
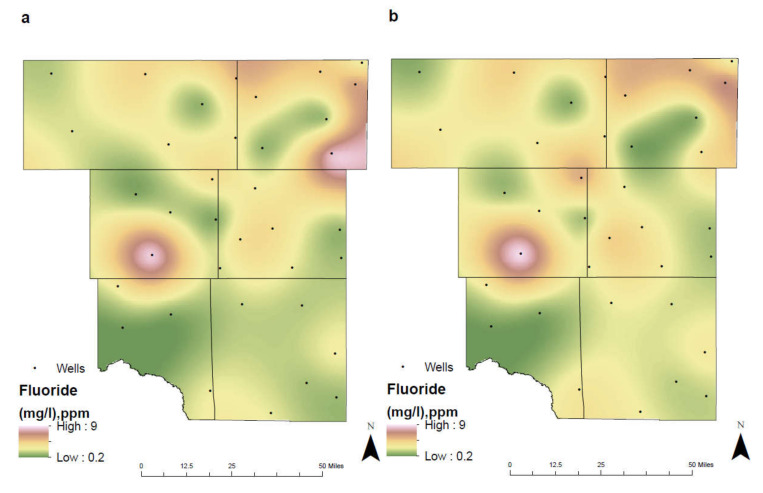
Spatial distribution of fluoride in the study are for (**a**) 1992–2005 and (**b**) 2006–2019.

**Figure 8 ijerph-17-04026-f008:**
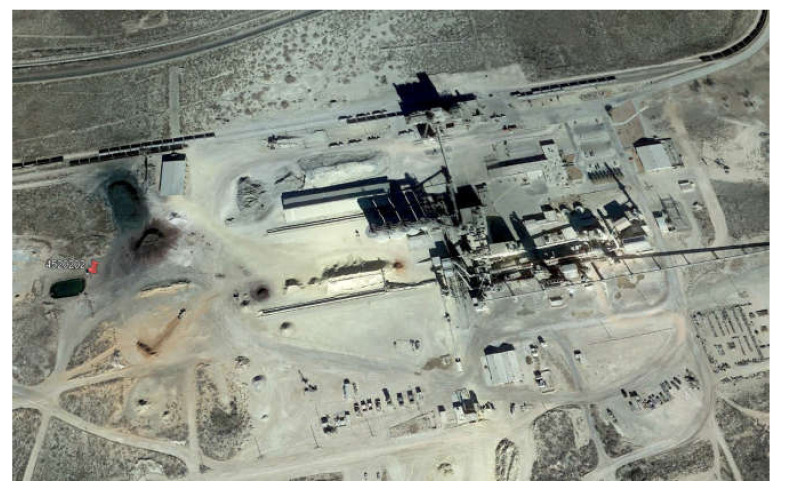
Location of well (ID: 4520202) adjacent to the cement production facility in southwest Ector County (image from Google Earth).

**Figure 9 ijerph-17-04026-f009:**
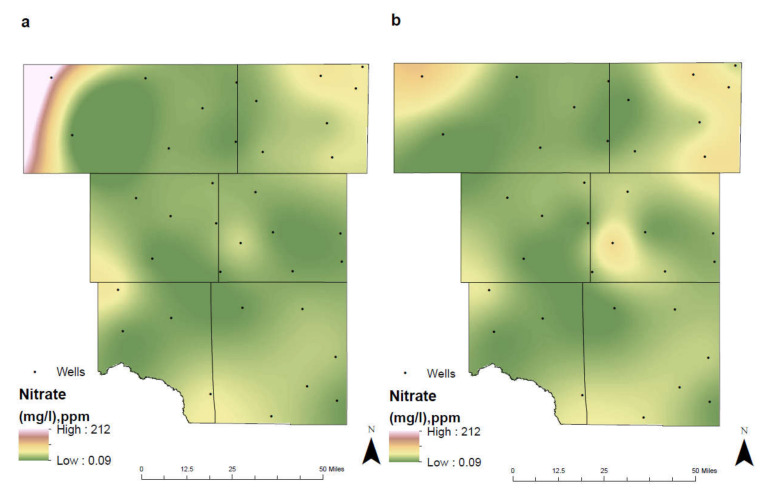
Spatial distribution of nitrate in the study area for (**a**) 1992–2005 and (**b**) 2006–2019.

**Figure 10 ijerph-17-04026-f010:**
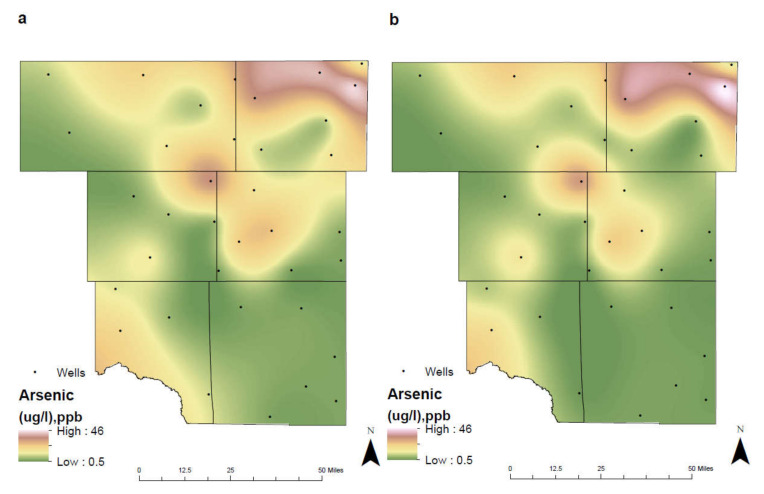
Spatial distribution of arsenic in the study are for (**a**) 1992–2005 and (**b**) 2006–2019.

**Figure 11 ijerph-17-04026-f011:**
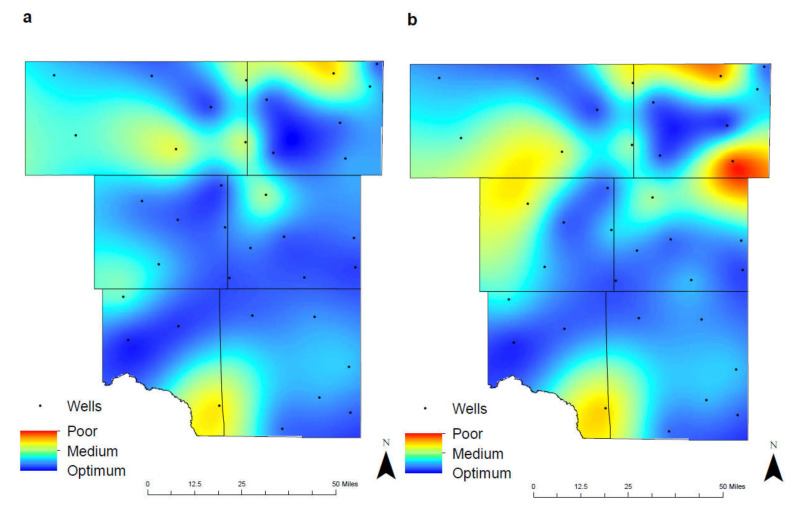
Spatial distribution of total groundwater quality from optimum to poor conditions in the study area for (**a**) 1992–2005 and (**b**) 2006–2019.

**Figure 12 ijerph-17-04026-f012:**
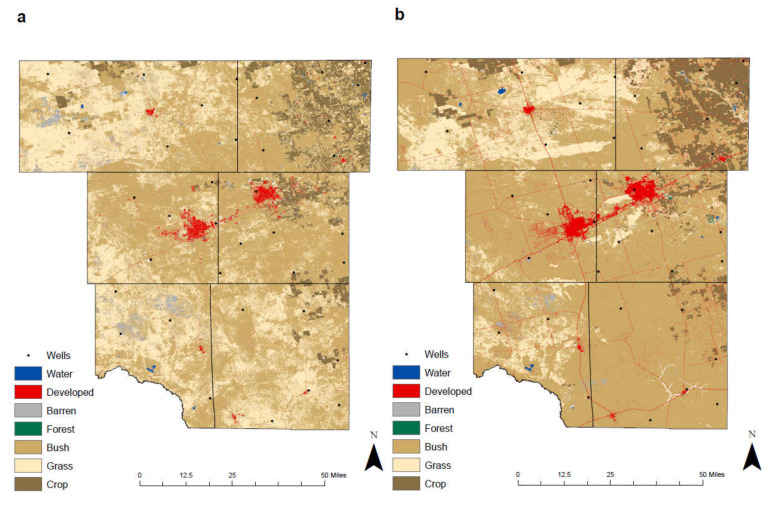
Land-cover maps of the study area; (**a**) NLCD 1992 and (**b**) NLCD 2011. NLCD: National Land Cover Dataset.

**Table 1 ijerph-17-04026-t001:** Summary of well data used in the study area (depth unit: feet. below land surface).

County	Well ID	Latitude	Longitude	Depth	Aquifer
Andrew	2750103	32.24111	−102.871	490	Pecos
	2753402	32.18833	−102.486	160	Ogallala
	2755103	32.21306	−102.219	1600	Dockum
	2746101	32.3475	−102.352	150	Ogallala
	2739405	32.45139	−102.215	215	Ogallala
	2736208	32.46889	−102.579	200	Ogallala
	2733202	32.47056	−102.954	120	Ogallala
Martin	2756401	32.17361	−102.111	70	Ogallala
	2850704	32.15028	−101.834	135	Ogallala
	2842705	32.28889	−101.855	122	Ogallala
	2739901	32.37764	−102.137	214	Ogallala
	2833303	32.47694	−101.88	125	Ogallala
	2835401	32.42833	−101.74	130	Ogallala
	2827705	32.51472	−101.714	133	Ogallala
Ector	4520202	31.74583	−102.552	1417	Dockum
	4506809	31.88722	−102.297	120	Edwards
	2762509	32.04667	−102.311	180	Edwards
	4505705	31.91519	−102.479	140	Edwards
	4504107	31.98806	−102.616	159	Edwards
Midland	4522604	31.69361	−102.28	225	Edwards
	4417409	31.69417	−101.992	255	Edwards
	4418214	31.73389	−101.796	176	Edwards
	4515505	31.8075	−102.199	180	Edwards
	2763901	32.01083	−102.141	90	Ogallala
	4410211	31.84722	−101.801	135	Edwards
	4516208	31.8525	−102.071	131	Edwards
Crane	4554501	31.20236	−102.32	200	Dockum
	4529704	31.50833	−102.477	80	Pecos
	4535506	31.45472	−102.669	190	Pecos
	4527203	31.62107	−102.689	94	Pecos
Upton	4564204	31.11333	−102.077	280	Edwards
	4450501	31.17639	−101.815	375	Edwards
	4449201	31.23278	−101.934	180	Edwards
	4442210	31.35139	−101.821	300	Edwards
	4425505	31.54389	−101.953	183	Edwards
	4531501	31.5488	−102.192	182	Edwards

**Table 2 ijerph-17-04026-t002:** Standards for groundwater quality by EPA and WHO (EPA: US environmental protection agency, WHO: world health organization, MCL: maximum contamination level).

Contaminant	MCL (mg/L)
pH	6.5–8.5
TDS	500
Chloride	250
Fluoride	4
Nitrate	10
Arsenic	0.01

**Table 3 ijerph-17-04026-t003:** Reclassification and land-cover description for the NLCD 1992 and 2011.

Reclassification	Land-Cover Description	
	NLCD 1992	NLCD 2011
Water	Open water	Open water
Developed	Low, medium, and high intensity residential	Developed low, medium, and high intensity
Barren	Bare rock/sand/clay	Bare land (rock/sand/clay)
Forest	Deciduous and evergreen forest	Deciduous, evergreen, and mixed forest
Bush	Shrubland	Shrub/scrub
Grass	Grassland/herbaceous	Grassland/herbaceous
Crop	Pasture/hay and row crops	Pasture/hay and cultivated crops

**Table 4 ijerph-17-04026-t004:** Groundwater conditions and the corresponding total concentrations.

Groundwater Conditions	Total Concentrations
Optimum	0–2600
Medium	2600–5200
Poor	5200—7800

**Table 5 ijerph-17-04026-t005:** The averages of six groundwater parameters in the study area for (a) 1992–2005 and (b) 2006–2019.

County	1992–2005						2006–2019					
	pH	TDS	Chloride	Fluoride	Nitrate	Arsenic	pH	TDS	Chloride	Fluoride	Nitrate	Arsenic
	(SU)	(mg/L)	(mg/L)	(mg/L)	(mg/L)	(µg/L)	(SU)	(mg/L)	(mg/L)	(mg/L)	(mg/L)	(µg/L)
Andrew	7.3	2347.5	551.8	2.7	36.2	12.3	7.3	2283.2	478.3	2.7	21.0	10.5
Martin	7.2	1540.9	400.3	3.9	42.7	21.8	7.1	2096.3	701.4	3.5	46.4	20.9
Ector	7.2	849.0	433.9	3.0	12.5	11.1	7.2	1693.2	563.3	5.5	13.2	12.3
Midland	7.2	1110.7	213.6	2.3	11.5	9.6	7.0	1292.3	278.0	2.5	30.3	8.6
Crane	7.3	2003.3	337.9	1.5	30.2	9.7	7.3	2005.2	401.7	1.7	24.5	7.2
Upton	7.1	1347.8	82.6	1.7	20.7	2.0	7.1	1356.6	87.9	2.1	24.1	1.7
Total	7.2	1533.2	336.7	2.5	25.6	11.1	7.2	1787.8	418.4	3.0	26.6	10.2

**Table 6 ijerph-17-04026-t006:** The area for each land-cover type in the study area (unit: km^2^).

	NLCD 1992	NLCD 2011	Changes
Water	0.5 (0.1%)	0.6 (0.1%)	0.1 (28.9%)
Developed	6.7 (1.2%)	18.5 (3.4%)	11.8 (176.1%)
Barren	13.1 (2.4%)	4.0 (0.7%)	−9.1 (−69.5%)
Forest	0.1 (0.0%)	0.3 (0.1%)	0.2 (277%)
Bush	295.4 (54.8%)	378.5 (70.2%)	83.2 (28.2%)
Grass	184.8 (34.3%)	91.6 (17.0%)	−93.2 (−50.4%)
Crop	38.5 (7.1%)	43.9 (8.1%)	5.4 (14.1%)
Others	-	1.6 (0.3%)	-
Total	539.0 (100%)	538.9 (100%)	-

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
