# Peer review of "Monitoring Environmental Parameters with Oil and Gas Developments in the Permian Basin, USA"

_ijerph, 2020, doi:10.3390/ijerph17114026_

Round 1

Reviewer 1 Report

This study has a potential to be cited by other studies and can be interesting for wide range of readers. The manuscript deals with the monitoring of environmental parameters with oil and gas developments in the Permian Basin, USA. The paper is very well structured. The introduction is informative, providing an overview of the scientific problem addressed and info from other relevant researches. The methodology is sound and presents satisfactory results thus, leading to sound interpretation. I think overall the paper is very good and, frankly, I hardly can find any flaws. 

For the sake of improvement, I would suggest the authors provide more details about the hydrogeological regime of the area (e.g. water level, hydraulic parameters, potential lateral crossflows, etc). I think the paper can be published in its current form.

Reviewer 2 Report

First, I wanted to thank the authors for their highly interesting study. I believe that it is well conducted and explained. However, I have a few minor comments:

Line 12:  In the abstract, “where environmental change could have potentially affected..” and not effected, maybe?

Line 31: “quality standards are met” and not meet, perhaps?

Line 41: “The three chemically derived contaminants that are not controlled by the EPA in this study are chloride, pH, and total dissolved solids (TDS).” Technically pH and TDS are not chemically derived contaminants. They are water properties. Please correct.

Line 46: Can you briefly summarize the major findings from the references 5-8? What are the effects of oil and gas production on groundwater quality and land use changes that other authors identified? Just to confirm what other studies found and strengthen your motivations.

Line 46: “Unconventional oil and gas production provide a..” would be “Unconventional oil and gas production provideS a..”

Line 50: “However, no studies have combined both parameters to identify how energy development and environmental change relate to groundwater quality.”. First, I would change “parameters” with factors, because the word parameter has a strict meaning in water resources. Second, what are these two parameters/factors? I believe that it is unclear. Are they land use changes and groundwater quality? I am not sure I understand this paragraph. Please revise.  

Line 134: Please summarize the methodology you adopted from Nas and Berktay [7]. Did they use spline interpolation? It is unclear.

Line 146: Throughout the manuscript, sometimes you use km2, sometimes meters squared. Please adopt a consistent format. For example, either you use km2 or squared kilometers. Either you use m2 or squared meters.

Line 233: “and must be disposed of properly to prevent degradation”, probably it would be: “and must be properly disposed to prevent degradation..”

Line 313: “Within alkaline environments (pH greater than 7), arsenic becomes mobilized allowing for increased concentrations to be observed. These environments promote the release of arsenic through the electrostatic repulsion of the negatively charged Fe oxides/hydroxides and arsenic compounds.” Please add a couple references to confirm that other studies agree with this statement (like you did in line 328).

Thank you.

Reviewer 3 Report

The study entitled” Monitoring Environmental Parameters with Oil and Gas Developments in the Permian Basin, USA” presents a study correlating groundwater qualities and environmental changes in the Permian Basin, Texas. Water quality parameters including pH, TDS, chloride, fluoride, nitrate, and arsenic from 36 wells in the study area were collected and analyzed. Two different periods of 1992-2005 and 2006-2019 were selected to compare changes in water quality. The study found that the groundwater quality decreased from 1992 to 2019 as population and urban growth began to develop in the Permian Basin because of anthropogenic activities such as chemical fertilizers, oil, and gas developments.  In general, the paper is well written and relevant, though there is a lack of deeper quantification analysis.  I would like to recommend the authors to use some statistical methods to quantify the relationships between different water quality parameters and their variations with environmental parameters. The current version is only based on figure comparison, which is not very scientifically convincing. There are several minor issuers:

1) Fig 4b has low resolution compared with Fig 4a.

2) Please check if you have used the correct fig for Fig5. b.

3) Both Fig.9a and 9b should have higher resolution.

4) Fig. 11a and 11b should have the same dimension.

Round 2

Reviewer 3 Report

The authors answered my questions. I am happy with the current manuscript.